# Fiber reinforced GelMA hydrogel to induce the regeneration of corneal stroma

Bin Kong[1,2,11], Yun Chen [3,11], Rui Liu[2,11], Xi Liu[4], Changyong Liu[5], Zengwu Shao[6], Liming Xiong[6], Xianning Liu[7,8], Wei Sun[1,9,10 ✉] & Shengli Mi[2,3 ✉]

Regeneration of corneal stroma has always been a challenge due to its sophisticated structure and keratocyte-fibroblast transformation. In this study, we fabricate grid poly (ε-caprolactone)-poly (ethylene glycol) microfibrous scaffold and infuse the scaffold with gelatin methacrylate (GelMA) hydrogel to obtain a 3 D fiber hydrogel construct; the fiber spacing is adjusted to fabricate optimal construct that simulates the stromal structure with properties most similar to the native cornea. The topological structure (3 D fiber hydrogel, 3 D GelMA hydrogel, and 2 D culture dish) and chemical factors (serum, ascorbic acid, insulin, and β-FGF) are examined to study their effects on the differentiation of limbal stromal stem cells to keratocytes or fibroblasts and the phenotype maintenance, in vitro and in vivo tissue regeneration. The results demonstrate that fiber hydrogel and serum-free media synergize to provide an optimal environment for the maintenance of keratocyte phenotype and the regeneration of damaged corneal stroma.

[1] Macromolecular Platforms for Translational Medicine and Bio-Manufacturing Laboratory, Tsinghua-Berkeley Shenzhen Institute, 518055 Shenzhen, P.R. China. [2] Biomanufacturing Engineering Laboratory, Tsinghua Shenzhen International Graduate School, 518055 Shenzhen, P.R. China. [3] Open FIESTA Center, Tsinghua Shenzhen International Graduate School, 518055 Shenzhen, P.R. China. [4] Beijing Children's Hospital, 100045 Beijing, P.R. China. [5] Additive Manufacturing Research Institute, College of Mechatronics and Control Engineering, Shenzhen University, 518060 Shenzhen, P.R. China. [6] Tongji Medical College, Huazhong University Science & Technology, 430022 Wuhan, P.R. China. [7] Shaanxi Institute of Ophthalmology, 710002 Xi'an, P.R. China. [8] Shaanxi Key Laboratory of Eye, 710002 Xi'an, P.R. China. [9] Department of Mechanical Engineering, Tsinghua University, 100084 Beijing, P.R. China. [10] Department of Mechanical Engineering and Mechanics, Drexel University, 19104 Philadelphia, PA, USA. [11]These authors contributed equally: Bin Kong, Yun Chen, Rui Liu. ✉email: sunwei@drexel.edu; mi.shengli@sz.tsinghua.edu.cn

The cornea locates at the outermost surface of the eye, and it plays an essential role in the visual system; it supplies two-thirds' of optical power, protects the intraocular structures and tissues, and refracts light onto the retina[1]. Injuries, bacterial and viral infections, and congenital and degenerative conditions may damage the function of the cornea, making corneal damage the second leading cause of blindness. More than 10 million individuals with diverse corneal disorders worldwide are reported every year[2]. Penetrating keratoplasty is the most commonly used grafting procedure to improve visual impairment from severe corneal diseases due to the high short-term success; however, the availability of donor corneal tissue cannot meet the global requirements[3]. Artificial keratoprostheses, animal decellularized corneal tissue (e.g., porcine cornea), and human amniotic membrane have also been used for the treatment of corneal disorders because of the shortage of donor cornea; however, these methods do not have high success ratios or approved use in tissue transplantation, despite that some methods are in current clinical practice[4]. For the reason above, it is essential and urgent to develop therapeutic alternatives to corneal transplantation, including cell-based therapy and bioengineered constructs.

Various bioengineering approaches have been attempted to fabricate corneal equivalence based on natural (e.g., collagen[1,4,5], gelatin[6], chitosan[7], silk[8], etc) or synthetic (e.g., poly (ethylene glycol) (PEG)[9], poly (ε-caprolactone) (PCL)[10], poly(lactic-co-glycolic acid) (PLGA)[1,11], poly-hydroxyethylmethacrylate (PHEMA)[12], etc) materials or the combination of natural and synthetic materials[1,4,13] by using the techniques of casting[14], hydrogel[15], 3D printing[16], electrospinning[17,18], and the combination of two or more of these processes[17]. Although these therapies and constructs have demonstrated acceptable mechanical properties and optical transmittance and can support corneal cells adhesion, migration, proliferation, and differentiation well, they fail to mimic the natural microenvironment of the native complex corneal tissue, and the most complicated part among the corneal tissue is the stroma. Corneal stroma, which constitutes 90% of the whole cornea, consists of orthogonally aligned collagen nanofibrous lamellae[1]. Collagen fibers of each lamella are parallel, tightly packed, and highly uniform in diameter. The transparency of stroma to light is attributed to the unique tight packing and uniform diameter of the collagen fibers, which in turn is regulated by glycosaminoglycan (GAG) and proteoglycans (PG) forming bridges between the collagen fibers. Between the lamellae, there are keratocytes with a dendritic phenotype that are quiescent in the normal situation and can express high levels of keratocan, ALDH3A1, and keratan sulfate. When the cornea is damaged or inflamed, the quiescent keratocytes can be activated, and transform into fibroblasts and myofibroblasts, which can secrete a large quantity of randomly distributed collagen fibers to repair the damage. However, the presence of regenerated random fibers can result in the scattering of incident light and thereby cause a deterioration of visual function due to the lack of fiber directionality[19]. Thus, in the regeneration of corneal tissue, the mimicry of the orthogonally aligned fibrous structure and the maintenance of the keratocyte phenotype are essential and critical.

Uniaxial aligned poly (ester urethane) urea (PEUU) sub-microfibers have been fabricated by electrospinning to mimic the aligned structure of stroma, and the human stromal stem cells seeded on the aligned fibers can express high levels of keratocyte-specific makers and secrete an aligned, dense collagenous matrix[18]. Orthogonally aligned poly-L-lactic acid (PLLA) sub-microfibers prepared by electrospinning also maintain the keratocyte phenotype[17]. However, the uniaxial or orthogonal fibers fabricated by electrospinning are only partly aligned due to the limitations of electrospinning in precisely controlling the network architecture. Moreover, the electrospun fibers are very dense, and the cells inoculated on the fiber surface cannot migrate into the inner

structure, resulting in the two-dimensional (2D) rather than three-dimensional (3D) cell culture. The dense electrospun fibers can also dramatically decrease the light transmittance of the fibrous scaffold, disqualifying it as a suitable corneal tissue replacement. Recently, near-field electrospinning or direct writing[20,21] has enabled the layer-by-layer assembly of the sub-micro-scale fibers and precise control of the network architecture, which may overcome these limitations, when compared with electrospun meshes[22], biotextiles[23], and 3D printing[24] constructs, which have been widely used as fibrous structure to engineer various tissues.

Hydrogels from natural materials[25,26] are also widely used in the construction of corneal tissue due to their excellent biocompatibility, intrinsic cell binding sites, 3D highly porous structure and highly aqueous environment. However, the poor mechanical stability of natural hydrogels can limit their application in tissue engineering. Hydrogels from synthetic materials have better mechanical strength, but worse biocompatibility[9]. Thus, to obtain a hydrogel with good biocompatibility and mechanical properties simultaneously, semi-synthetic and chemically functionalized hydrogels (e.g. gelatin methacrylate (GelMA)[27–29]) emerged. Recently, the combination of hydrogels and fibrous scaffold has been used to engineer fiber-reinforced hydrogels[30,31] applied in the regeneration of soft tissues, including cartilage[32–36], heart valve[23,37,38], tendon[39], and muscle[40]. The fibrous structure of the fiber hydrogel construct can resemble and simulate the biological fibers of the native soft tissues and supply the hierarchical and morphological cues to induce the specific growth and differentiation of cells. Besides, the fibrous scaffolds can enhance the mechanical performance of the hybrid construct by transferring the most load from the hydrogels to the fibers, and the mechanical strength can be easily regulated by changing the materials and processing parameters to engineer a construct with structure and mechanical properties most similar to the targeted soft tissues, especially for the fibrous scaffolds fabricated from the near field electrospinning. The hydrogel of the fiber hydrogel construct can resemble the matrix and supply a real 3D aqueous and porous environment for the cells to adhere, migrate, proliferate, and differentiate. And the semi-synthetic hydrogels can also supply a highly regulatory and controllable chemical modifications for specific cells and tissues.

Thus, in this study, to mimic the native corneal stroma structure, we precisely fabricate orthogonally aligned poly (ε-caprolactone)-poly (ethylene glycol) (PECL) sub-micro fibers, which are perfused by the GelMA hydrogel to form a 3D fiber-reinforced hydrogel. We determine the optimal fiber hydrogel to be used as a corneal equivalent by studying the effects of direct writing different fiber topological structures on the mechanical strength, light transmittance, and mass swelling ratio. To maintain the keratocyte phenotype, we study the influences of topological structures (2D tissue culture plates (TCP), 3D GelMA hydrogel, and fiber hydrogel) and chemical factors (serum, insulin, β-FGF, and ascorbic acid) on the differentiation of limbal stromal stem cells (LSSCs) by measuring the expression of keratocyte-specific and fibroblast-specific proteins and genes. Finally, we demonstrate that the designed fiber hydrogel can induce the regeneration of damaged corneal stroma in vitro and in vivo.

## Results
The schematic diagram of the experimental process is shown in Supplementary Fig. 1.

**PECL synthesis and fabrication of PECL microfibers.** PECL copolymers were synthesized successfully by the open-ring polymerization of ε-CL monomer initiated by bio-functional HO-PEG-OH, which can significantly improve the hydrophilicity of PCL, shown in Supplementary Figs. 2–5.

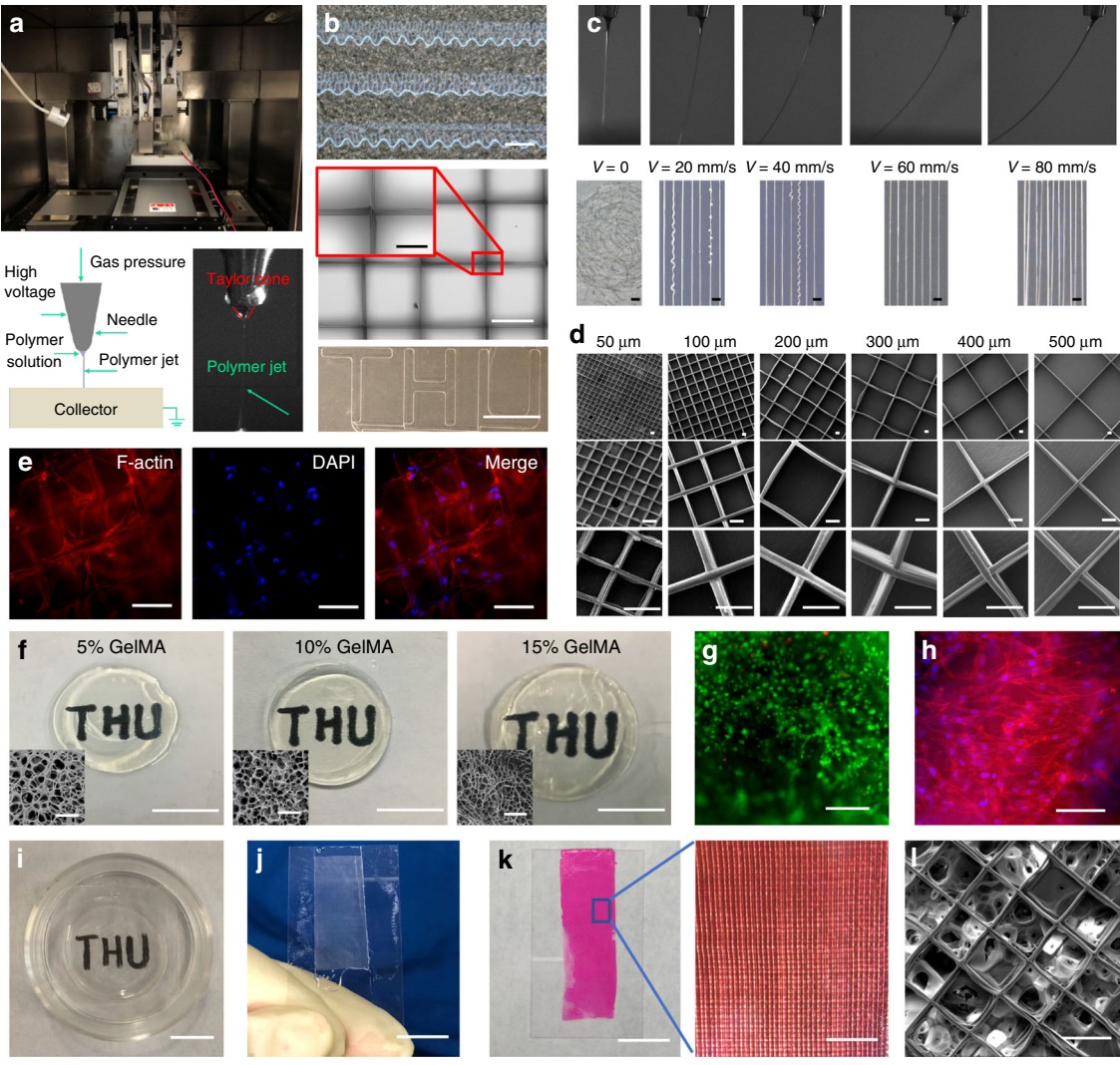

**Fig. 1 Fabrication of characterization of the fiber hydrogel. a** The custom-made direct writing device, a schematic representation of direct writing, and a polymer jet being ejected from a Taylor cone. **b** Images of the obtained curly lines (Scale bar is 500 μm), grids (Scale bar is 500 μm and the enlarged image is 150 μm), straight lines, and a specifically designed pattern (Scale bar is 10 mm). **c** Images of stable jet deposition at different the moving speeds of the substrate in the specific conditions. The scale bar is 100 μm. **d** SEM images of the grid scaffolds with fiber spacing of 50–500 μm under different magnification. The scale bar is 50 μm. **e** LSSCs inoculated on the grid scaffolds can be induced to grow along the fiber direction, cells were stained with phalloidin (red) and DAPI (blue). The scale bar is 100 μm. **f** The macroscopic and microscopic morphology of the formed GelMA hydrogels with 5%, 10%, and 15% concentrations. The scale bar for macroscopic images is 10 mm, for the SEM images is 80 μm. **g** The viability of LSSCs within the 5% GelMA hydrogel. Live cells are visualized with green, and dead cells appear red. The scale bar is 500 μm. **h** The cytoskeleton (red) and DAPI (blue) staining of LSSCs within the 5% GelMA hydrogel. The scale bar is 100 μm. **i–l** The fabricated 100 μm spacing grid fiber-reinforced GelMA hydrogel. The scale bar in **i–k** is 10 mm, the enlarged image in **k** is 1 mm, **l** is 100 μm. The staining in **e**, **g**, and **h** were repeated three times independently with similar results.

PECL copolymer with a MW of 8 WDa was used to fabricate sub-microfibers successfully by direct writing. Figure 1a shows the actual and schematic representation of the direct writing device, and a polymer jet was ejected from the Taylor cone. By using this device, various forms of fiber patterns can be achieved, such as the curly lines, grids, straight lines, and specifically designed patterns (e.g., THU) shown in Fig. 1b. In the direct writing process, the jet formation speed and the collector motion velocity played an essential role in controlling the deposited fibers' morphology. We studied the effects of collector velocity on the fiber wagging and morphology under the following conditions: voltage 3.6 kV, distance 3 mm, and gas pressure 20 kPa. With increased velocity, the jet wagging angle increased, and the chance of presence of depositing curly fibers decreased, as shown in Fig. 1c. However, if the velocity is too large, the oversized wagging angle could result in the deviation of the fiber depositing

trace from the designed path. A velocity of 60 mm/s was ultimately chosen to fabricate grid scaffolds with fiber spacings ranging from 50 to 500 μm. The morphology of fibers and deposition accuracy were observed by SEM at the magnifications of ×300, ×1000, and ×2500 (Fig. 1d). The fibers showed high deposition accuracy, and the average diameter was ~5 μm. The LSSCs inoculated on the grid scaffold can adhere to the fibers and spread along the orthogonally aligned structure (Fig. 1e).

**Preparation of GelMA and fiber hydrogel construct.** GelMA hydrogels at 5%, 10%, and 15% concentrations (Fig. 1f) were prepared to assess the cell viability and spreading within hydrogels of different stiffness, and the high viability and degree of spreading were only observed within 5% GelMA, as shown in Fig. 1g, h. This result may be due to the larger pore size of the 5%

hydrogel compared to the higher concentrations of GelMA. The fiber hydrogel was fabricated by infusing the GelMA solution into the mold with the 50–500 μm grid scaffold and named 50G, 100G, 200G, 300G, 400G, and 500G, respectively. The 100G construct is shown in Fig. 1i–l with high light transmittance. In Fig. 1k, the GelMA solution was pre-stained with red dye to confirm the formation of the hybrid fiber hydrogel. The SEM image in Fig. 1l can more clearly show the microscopic structure of the fiber hydrogel.

**Physiochemical characterization of fiber hydrogel.** The tensile and compressive strength, the light transmittance, and the swelling ratio of the 50G, 100G, 200G, 300G, 400G, and 500G constructs were measured to study the effects of fiber spacings on these properties and to verify the optimal spacing for produing a construct with properties most similar to the native corneal tissue. The tensile strain–stress curves and elongation at break of fiber hydrogels and acellular cornea after immersing into PBS for 1 h (day 0) are shown in Fig. 2a, b; a large elongation at near constant stress were observed from the strain–stress curves of the fiber hydrogels after the materials yielded, which is similar to that of a typical flexible plastic[41], such as polyethylene, polyvinyl chloride, and polytetrafluoroethylene; and with increased fiber spacing, the maximum tensile stress and elongation at break both decreased. The 50G and 100G constructs can attain the tensile strength requirements of the native cornea (3–5 MPa)[42], with the maximum tensile strengths of $3.79 \pm 0.51$ and $3.47 \pm 0.43$ MPa, (The data were calculated as mean ± standard deviation (SD) ($n = 3$)), respectively. All constructs had high elongation at break, indicating that the fiber hydrogels had high ductility much greater than that of the native cornea. The maximum tensile strength of fiber hydrogels after immersing into PBS for 7, 14, and 28 days were also measured to verify the long-term mechanical stability of the constructs, as shown in Fig. 2c. With increased immersion time, the tensile strength of all constructs decreased to a certain degree. However, the 50 and 100 G constructs can still meet the requirements of the native cornea with the tensile strengths of $3.42 \pm 0.46$ and $2.93 \pm 0.36$ MPa after 28 days. Figure 2d shows the compressive modulus of pure GelMA hydrogel, fiber hydrogels constructs, and the native cornea. The addition of grid fibers can dramatically enhance the compressive modulus of GelMA, and the smaller the fiber spacing, the larger the modulus. The effect trend is consistent with previous research results[34,35]. Although the fibrous scaffolds alone can bear limited compressive strength due to the intrinsic properties, including fiber delamination and buckling, for the fiber hydrogels, the compression would cause the stretch of hydrogels within the grid fibrous scaffold, thus, the longitudinal compressive force would transfer to transverse tensile force to the fibrous scaffold. Due to the excellent tensile strength of the fibrous scaffolds, the fiber hydrogels can also have better compressive strength[35]. Except for 400G and 500G, other fiber-reinforced constructs all have the compressive modulus similar to or greater than the native corneal tissue.

The light transmittance of the fiber hydrogels and native cornea were determined by using a microplate reader at the wavelength ranging from 400 to 750 nm after immersing into PBS for 1 h (day 0) (Fig. 2e). The increase of fiber spacing resulted in the increased light transmittance of the constructs, especially when the fiber spacing was greater than 200 μm. Figure 2f shows the transmittance histogram of the fiber hydrogels at the wavelength of 500 nm after immersing into PBS for 0, 7, 14, and 28 days. The fiber spacing effects on day 7 and after were the same as on day 0. Moreover, for a single fiber hydrogel, the light

transmittance increased over time, with an apparent increase on day 7 and minor increase thereafter.

Swelling ability of hydrogels, which is influenced by hydrogel pore size, can indicate the degree of hydrophilicity and is an essential feature of the cornea. The measured mass swelling ration of the native cornea and pure GelMA hydrogel was $14.5 \pm 0.59$ and $16.7 \pm 0.51$ (The data were calculated as mean ± SD ($n = 3$)), respectively. From the histogram shown in Supplementary Fig. 6, the addition of grid fibers can reduce the swelling ratio of GelMA hydrogel, and the swelling ratio increases with increased fiber spacing. This result may because the micro-scale grid pore inhibits the increase of volume that was initiated by the permeation of water in the normal hydrogels. Compared with the remaining constructs, the swelling ratios of the 100G and 200G construct were more similar to the native cornea at $13.9 \pm 0.49$ and $14.6 \pm 0.25$, respectively.

Considering the mechanical properties, light transmittance, and mass swelling of these constructs, the 100G construct with tensile strength $3.47 \pm 0.43$ MPa, compressive modulus $121.3 \pm 10.2$ KPa (The data were calculated as mean ± SD ($n = 3$)), light transmittance of $74.7 \pm 1.45\%$ at 500 nm wavelength (The data were calculated as mean ± SD ($n = 3$)) and a mess swelling ratio of $13.9 \pm 0.49$ was most similar to the native corneal tissue. The following in vitro and in vivo experiments were all based on the 100G construct.

**Topological and chemical effects on the keratocytes.** LSSCs inoculated on 2D TCP, 3D GelMA hydrogel, and 100G constructs in SC media and SF media were used to study the effects of the three topological structures and chemical factors (serum, insulin, β-FGF, and ascorbic acid) on the differentiation of LSSCs to keratocytes and the maintenance of the keratocyte phenotype. After culturing for 7 days, the expression of Vimentin[43] (specifically expressed in fibroblasts) and ALDH3A1[1,44] (specifically expressed in keratocytes) in SC and SF media, respectively, were investigated by immunofluorescence (IF) staining. As shown in Fig. 3a, b, in SC media, cells inoculated on 2D and 3D substrates both expressed the Vimentin, but Vimentin was minimally expressed on the 100G construct. In SF media, cells inoculated on all constructs expressed ALDH3A1, indicating that in serum containing media, the keratocytes on 2D and 3D substrates can easily transform into fibroblast-like cells; however, the 100G construct can inhibit the transformation. In serum-free media supplied with ascorbic acid, β-FGF, and insulin, cells can maintain the keratocyte phenotype on all constructs, especially on 100G.

To further verify the topological and chemical effects, the gene expression of the keratocytes specific markers Keratocan (KERA)[45,46], ALDH3A1 (ALDH), Aquaporin 1 (AQP1)[45], and the fibroblasts specific marker Thymocyte antigen 1 (THY1)[17,47] after culturing for 2 weeks were assessed by qPCR. From the results shown in Fig. 3c, d, in SC media, the expression of KERA and ALDH were up-regulated on both 3D and 100G; expression was higher on 100G, but the expression of AQP1 showed no big difference on 3D or 100G. The expression of THY1 was down-regulated significantly in both 3D and 100G and to a greater extent in 100G; in SF media, there was no significant difference in the expression of KERA, ALDH in 3D, but in 100G, the expression of these genes was significantly up-regulated. The expression of THY1 were also both down-regulated in 3D and 100G, and 100G was significantly down-regulated. As shown in Fig. 3e–h, the expression of KERA, ALDH, and AQP1 in SF media were up-regulated in all constructs, and the expression of THY1 in SF media were down-regulated in 3D and 100G,

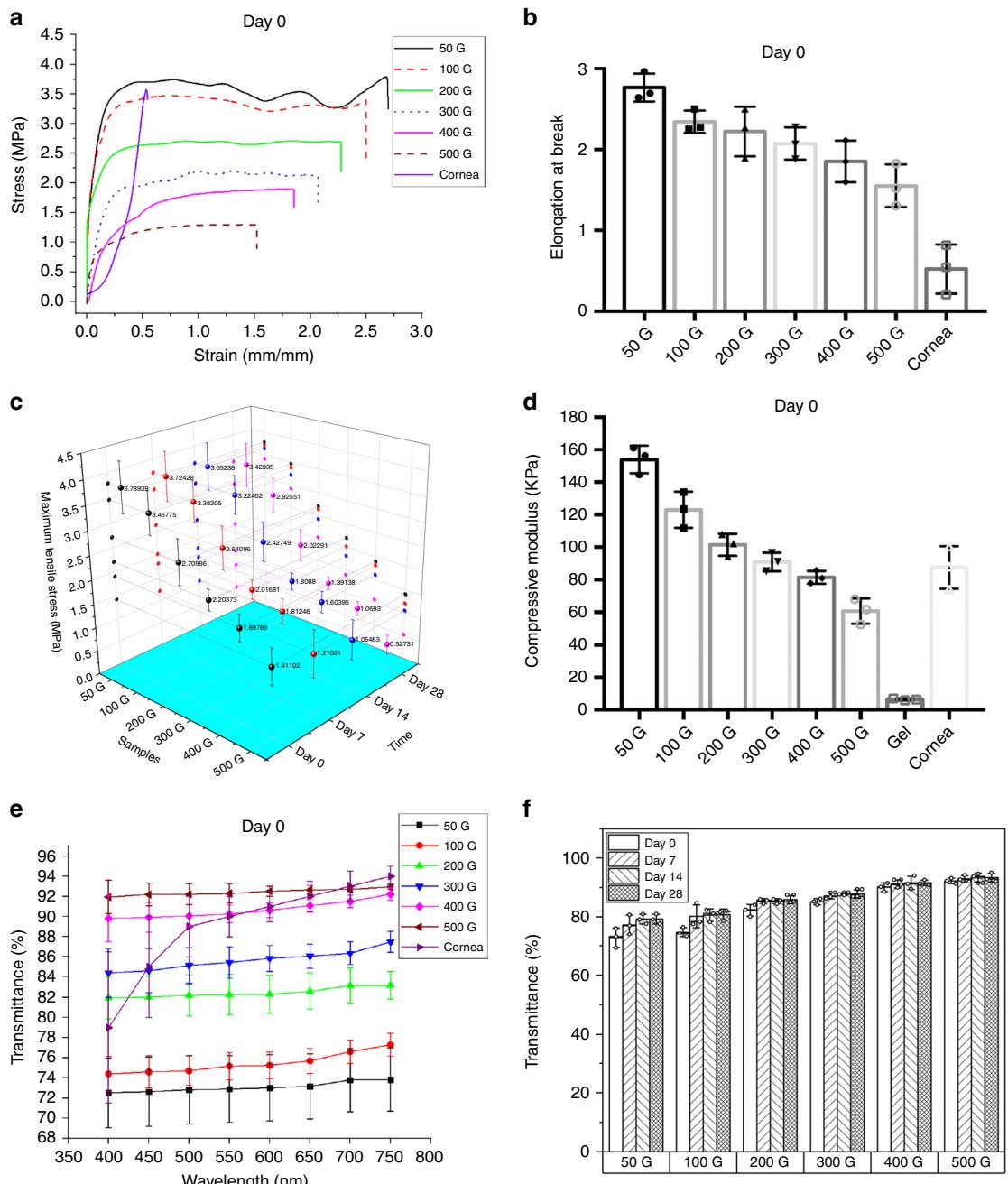

**Fig. 2 Fiber hydrogel properties. a** Average strain–stress curves of 50G, 100G, 200G, 300G, 400G, and 500G constructs and the acellular cornea after incubating in PBS for 1 h. **b** Histogram of strain at break of the samples in **a**. **c** Maximum tensile stress histogram of the samples in A after incubating in PBS for 0, 7, 14, and 28 days. **d** Histogram of compressive modulus of 50G, 100G, 200G, 300G, 400G, and 500G constructs, the acellular cornea, and the pure GelMA hydrogel. **e** The average transmittance spectra of 50G, 100G, 200G, 300G, 400G, and 500G constructs and the acellular cornea after incubating in PBS for 1 h (day 0). **f** Transmittance histogram of 50G, 100G, 200G, 300G, 400G, and 500G construct after 0, 7, 14, and 28 days of immersion in PBS at the 500 nm wavelengths. All the results were calculated as mean ± standard deviation (SD) (n = 3, biologically independent samples). Source data are available in the Source Data file.

significantly in 2D. These results indicated that on a 2D substrate, the serum can induce the keratocytes to transform into fibroblasts, whereas serum-free media prohibited the transformation to a certain degree. However, in the 100G fiber hydrogel, the keratocytes can maintain their phenotype, even with the presence of serum. Additionally, the synergistic effects of 100G and serum-free media can provide the most suitable topographical and chemical environment for the maintenance of the keratocyte phenotype.

After culturing for 4 weeks, the LSSCs inoculated on 2D, 3D, and 100G in SF media were examined to study the expression of collagen VI using immunostaining and Sirius staining, a unique feature of the ECM of corneal stromal tissue[48], which is largely expressed by keratocytes. Because the 3D hydrogel would contract a lot initiated by the reaction of the encapsulated keratocytes after culturing for 4 weeks, the staining result was not representative. However, the 100G construct does not contract and been affected by the keratocytes due to the reinforcement of grid fibers. From

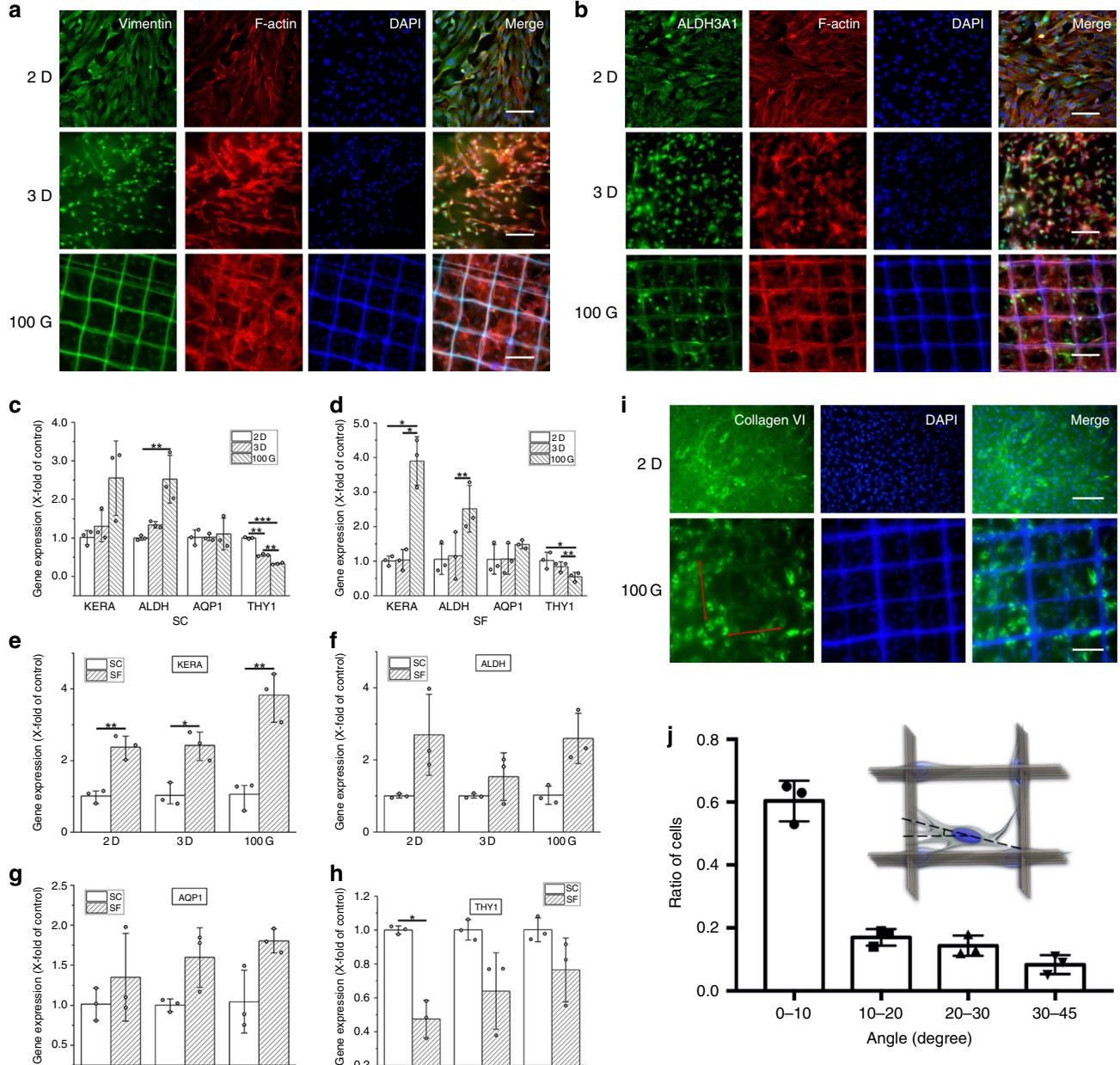

**Fig. 3 Effects of topological structure and chemical factors on keratocytes. a** Vimentin expression and cytoskeleton staining of LSSCs in SC media on 2D TCPs, 3D GelMA, and the 100G construct after culturing for 7 days. Images show the fluorescent staining of Vimentin (green), phalloidin (red), and nuclei (blue). The scale bar is 100 um. **b** ALDH3A1 expression and cytoskeleton staining of LSSCs in SF media on 2D TCPs, 3D GelMA, and the 100G construct after culturing for 7 days. Images show fluorescent staining of ALDH3A1 (green), phalloidin (red), and nuclei (blue). The scale bar is 100 um. **c–h** The expression of KERATOCAN, ALDH3A1, AQP1, and THY1 in LSSCs cultured on 2D TCPs, 3D GelMA, and 100G construct ($n = 3$, biologically independent samples) were quantified by qPCR after cultured for 2 weeks in SC media or SF media; quantification was normalized by the β-actin signal. Data are expressed as the mean ± SD. *$P < 0.05$; **$P < 0.01$; ***$P < 0.001$; Unpaired two-tailed student's $t$ test. **i** Immunofluorescent staining of the collagen VI (green) secreted by LSSCs inoculated on 2D TCPs or in 100G construct in SF media after culturing for 4 weeks. Nuclei were stained with DAPI (blue). The scale bar is 100 μm. **j** The cell alignment degree along the orthogonally aligned fibers on the 100 G construct ($n = 3$, biologically independent samples) after 4 weeks culture under the SF media. The data were calculated as mean ± SD. The staining in **a**, **b**, and **i** were repeated three times independently with similar results. Source data are available in the Source Data file.

Figs. 3i and S7, collagen VI was abundantly expressed in both 2D and 100G. The collagen secreted on 2D was random, demonstrating no alignments, however, the collagen secreted on 100G exhibited orthogonal alignment along the grid fibers (the red lines in Fig. 3i showed the direction of the fibers), which can be demonstrated by the cell alignment degree along the fibers shown in Fig. 3j. We measured the degrees of 300 nuclei of cells in three

100G construct after 4 weeks culture under the SF media and calculated the percentage of the cells within a range of degree, including 0–10°, 10–20°, 20–30°, and 30–45°, and we assumed that the cells within 0–10° were aligned along the fibers. These results indicated that the keratocytes can maintain their phenotype in SF media after culturing for 4 weeks, and the grid scaffold can induce the secretion of orthogonally aligned keratocyte-specific ECM.

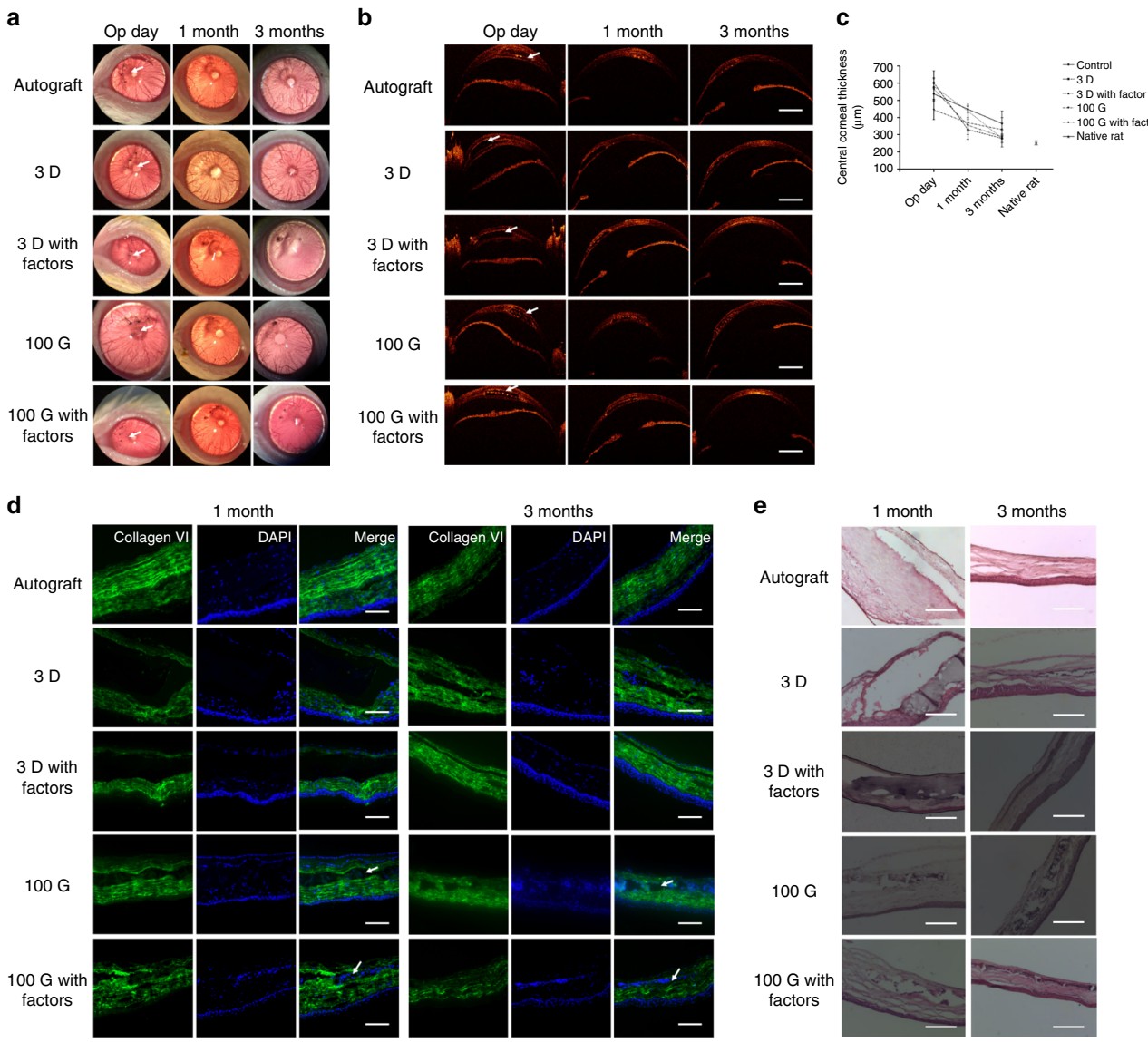

**Fig. 4 Fiber hydrogel engraftment and stromal matrix synthesis and regeneration in rat cornea in vivo. a** Images taken by slit lamp after operation, 1 and 3 months transplantation of the autograft, 3D GelMA hydrogel with or without chemical factors, 100G fiber hydrogel with and without factors. The white arrows indicate the position of the grafts. **b** The OCT images of the same corneas in **a** postoperatively. The white arrows indicate the position of the grafts. The scale bar is 1000 μm. **c** The thickness of the central corneas by measuring corneas (n = 3, biologically independent samples) in each group and each time point. Data were calculated as mean ± SD. **d** On 1 and 3 months, frozen sections of the transplanted corneas were immunostained with collagen type VI antibodies (green). In all images, nuclei were co-stained with DAPI (blue), the scale bar is 100 μm. **e** H&E staining of the transplanted corneas on 1 and 3 months, the scale bar is 100 μm. The experiments in **a**, **b**, **d**, and **e** were repeated three times independently with similar results. Source data are available in the Source Data file.

**Rat intrastromal keratoplasty and evaluation**. The autograft, 3D GelMA hydrogel, 3D GelMA hydrogel with chemical factor, 100G fiber hydrogel and 100G fiber hydrogel with chemical factors were transplanted into corneas of 30 rats using the intrastromal keratoplasty. The white arrows in Fig. 4a indicated the positions of these transplanted grafts. One week after operation, severe inflammation was observed in a rat with 3D hydrogel, and a rat with 100G fiber hydrogel. They were pre-sacrificed because the symptom did not subside or get better after treatment with antibiotics. Besides, four rats died accidentally 1 and 2 months after surgical operation, and we re-transplanted the grafts. And the total successful rate of the transplantation was over 60%. One month postoperatively, 4 of 30 corneas were observed with neovascularization (two in 3D hydrogel, one in 3D hydrogel with

factors and one in 100G fiber hydrogel), and the symptom disappeared after 3 months without any treatment. From the images taken by slit lamp shown in Fig. 4a, 3 months postoperatively, the transplanted corneas all showed high light transmittance. From the OCT images shown in Fig. 4b, we can clearly observe the position of the transplanted grafts (white arrow) and the dots in the 100G with or without factors indicated the PECL fibers. Immediately after operation, all corneas showed apparent edema and significant increase of the thickness (Fig. 4c). One month after operation, the edema subsided and the thickness of all corneas have decreased, with anterior curvature following that of the posterior cornea. Three months after operation, the thickness of all corneas continually decreased but still had a little larger than that of the native rats. With the increase of time, the area of

the blank occupied by the grafts all decreased, indicating that the degradation of the scaffolds and the regeneration of the stroma.

1 month after operation, histological H&E staining and collagen VI immunofluorescence staining (Fig. 4d, e) both revealed that the graft with factors can induce the regeneration of the stroma better than the same graft without factors. And the graft with fibers can induce the regeneration of the stroma better than the same graft without fibers. The white arrows in 100G and 100G with factors indicated the PECL fibers, because the fibers cannot be fully degraded within 3 months. However, when comparing the 100G and 100G with factors, we can find out that the occupied positions by fibers in 100G with factors was smaller than that in the 100G, also indicating that the factors may accelerate the degradation of the fibers by stimulating the regenerated keratocytes to secrete the degrading enzyme. Three months after operation, the influence of the factors and fibers on the regeneration of the stroma was same with that in 1 month, and in the 3D and 3D with factors, the stroma were not fully regenerated. However, 100G and 100G with factors have fully induced the regeneration of the stromal tissue with normal morphology except for the positions occupied by the undegraded fibers. All results indicated that the synergistic effects of 100G and chemical factors can provide the most suitable topographical and chemical environment to induce the regeneration of the corneal stroma in vivo.

## Discussion

Corneal stromal regeneration is the most difficult part of whole corneal repair due to the sophisticated orthogonally aligned fibrous lamellar structure and the small number of keratocytes. Moreover, the keratocytes can easily transform into fibroblasts and myofibroblasts during the restoration of injured cornea or in vitro culture. Thus, to obtain large numbers of keratocytes and meanwhile maintain the keratocyte phenotype, inhibiting the transformation is another difficulty need to be addressed[49]. Many researchers have used the technique of electrospinning to fabricate aligned fibers that simulate the native stromal structure; however, due to the bending instability of the jet formed when the electrical charge repulsive force overcomes the surface tension of the electrospinning solution during the electrostatic repulsion, the control over the fabricated fibers is very poor[50]. Although some methods have been developed to obtain aligned fibers, these fibers were only partially aligned[17,18]. Thus, we cannot precisely control the pattern of the obtained fibers by electrospinning. Moreover, the electrospun fibers are very dense, resulting in the low light transmittance, which makes it not applicable in the transparent cornea. Recently, the near-field electrospinning or the direct writing technique was developed from the conventional electrospinning to improve the control over fibers[20,21]. Because of the millimeter-scale distance between the nozzle and collector, the bending instability issue of electrospinning can be avoided. The most widespread materials used in melting direct writing is PCL due to its low melting temperature and stability. Although PCL is biocompatible, its hydrophobicity can limit its applications in tissue engineering. To increase the hydrophilicity of PCL, in this study, we successfully synthesized the PEG and PCL copolymer PECL. By the addition of hydrophilic PEG, the hydrophilicity of PCL can be improved significantly, as demonstrated by the contact angle changing from degree of 68.4 ± 4.2 to 49.1 ± 2.7. At the 61 °C melting temperature, the synthesized PECL copolymer was successfully fabricated into a sub-microfibrous scaffold by direct writing for the first time, to the best of our knowledge. With the high accuracy X–Y moving platform and well-studied processing parameters, the fibers with an average of 5 μm diameter and high deposition accuracy were fabricated in arbitrary patterns.

Grid-like PECL microfibrous scaffolds were fabricated to simulate the orthogonally aligned stromal lamellae, and infused with a GelMA hydrogel to prepare the fiber hydrogel firstly used in corneal tissue engineering. It should be clarified that the synthesized PECL was not medical grade, thus the fiber hydrogel cannot be used clinically. The combination of hydrogel and a fibrous grid scaffold can reinforce the strength of the hydrogel to address the common problem that most hydrogels are not suitable in surgical sutures. Additionally, the fibrous scaffold within can provide a topological cue to cells that the conventional tissue-engineered corneal hydrogel cannot attain. Moreover, compared with an electrospun fibers-reinforced hydrogel, our fiber hydrogel can provide higher light transmittance and allow the 3D culture of cells due to the micro-scale fiber spacing instead of the nano-scale pore size of electrospun fibrous scaffold that inhibits the growth of cells into the inner scaffold. For the designed fiber hydrogel construct, the fiber spacing is a critical factor that affects the properties of the whole construct; we confirmed that the GelMA pre-gel solution with the concentration of 5% w/v allowed good viability and spreading for encapsulated LSSCs. When preparing a scaffold used in tissue-engineered cornea, the mechanical strength, light transmittance, and swelling ratio should be considered as a designing criteria, because these properties are essential to corneal tissue. Thus, we designed fibrous grid scaffolds with six different fiber spacings, ranging from 50 to 500 μm to study the relationship between the fiber spacing and construct properties and to identify the optimal fiber spacing. From the results shown in Fig. 2 and Supplementary Fig. 3, we easily discerned that the maximum tensile stress, elongation at break, and compressive modulus of the constructs were inversely proportional to the fiber spacing, and the light transmittance and mass swelling ratio were proportional to the fiber spacing. These phenomena can be consistently interpreted as fiber numbers effects. When the shape and area of samples were the same for one measurement, the smaller fiber spacing means the more fibers existing in the construct. With more fibers, the construct will undoubtedly have better mechanical properties, because the fibrous scaffold supplies most of the mechanical force in the construct. It should be recognized that by reinforcing with orthogonally aligned fibers, we can significantly improve the maximum tensile strength of the fiber hydrogels and attain the strength requirement of the native corneal tissue. But from Fig. 2a and previous research results[41], we can find out that the native soft tissues exhibit a J-shaped strain–stress curves with three distinct phases, which is different from that of the designed fiber hydrogels. The difference may because the straight fibers were already taut, and the stress would rapidly increase without a gradual increase when the stretch began. Onur Bas et al used near-field electrospinning to fabricate curved fibers, and have demonstrated that the reinforced hydrogels with curved fibers successfully displayed a J-shaped strain–stress curves[41]. And future effort can be applied to the design of the fiber patterns to make a construct with mechanical properties most similar to the native corneal tissues. For the light transmittance measurement, because opaque fibers can reduce the light transmittance, more fibers reduce the whole construct transparency. For the swelling ratio measurement, more fibers mean that more chambers which are formed by the X axial fibers and Y axial fibers. More chambers can constrain the volume increase of the hydrogel within the chambers, resulting in the reduction of the swelling ratio.

Except for the structural simulation, the keratocyte source and the maintenance of its phenotype are additional challenges as mentioned above. In this study, a large number of LSSCs, which can differentiate into keratocytes, were obtained from the rat limbal, and these stem cells can proliferate rapidly in vitro on 2D culture dish and can be sub-cultured in specific stem cell media.

Previous studies have indicated that the aligned fibers can induce the LSSCs and keratocytes to grow along the orientation of the fibers and secrete an aligned ECM, whereas random fibers or casting membrane cannot induce such alignment. In our study, the encapsulation of the orthogonally aligned microfibers within the GelMA hydrogel can clearly affect the cell organization and morphology in the 100G construct, as demonstrated by the cytoskeleton staining. Cells on the 100G construct can grow along the two directions of fibers and distribute on different layers; additionally, the LSSCs on the 100G after culturing for 7 days tend to be dendritic, the typical morphology of keratocytes[51]. Chemical cues such as serum which has been verified that can promote the keratocytes to differentiate into fibroblast. In serum-free media, the supplementation is needed to avoid the cell apoptosis initiated by the removal of serum and meanwhile to promote the proliferation of cells without their transformation into fibroblasts[52]. The supplementation widely used includes insulin, which has been demonstrated to maintain the keratocyte viability and phenotype; β-FGF, which can stimulate cell proliferation and promote KERA and ALDH expression[46]; ascorbic acid, which has been used to induce the stromal cells to maintain a physiological morphology and spatial distribution of keratocytes and promote the synthesis of collagen I, V, and VI fibers[53]; and insulin, which can stimulate the synthesis of collagen without affecting the accumulation of lumican and KERA[54,55]. In our study, we chose the combination of insulin, β-FGF, and ascorbic acid as supplements in the serum-free SF media, and the results demonstrated that these factors can keep the keratocyte-like phenotype and promote the keratocyte-specific gene and protein expression (ALDH, KERA, AQP1), as shown in the IF staining, and qPCR data. Moreover, these factors can promote the secretion of the keratocyte-specific ECM collagen VI. We also studied the synergistic effects of topological and chemical factors and demonstrated that the topological grid microfibers can inhabit the differentiation of keratocytes into fibroblasts even in the serum containing media, and the combination of the grid microfiber and chemical factors without serum can attain the optimal qualification which can maintain the keratocyte phenotype and induce the orthogonally aligned ECM secretion similar to the native stromal tissue in vitro and in vivo.

## Methods

**Synthesis and characterization of PECL copolymer.** ε-CL (Sigma-Aldrich) was purified by reduced pressure distillation prior to polymerization. Dried polymeric tubing containing a specific quantity of PEG (Mw 10 kDa, purchased from Xiamen Sinopeg biotech) was prepared, and a pre-weighed volume of caprolactone monomer was added into the tubing before the addition of 100 μl 5% Sn (Oct)₂ (Sigma-Aldrich). The tubing was placed into an oil bath at 130 °C for 18 h after evacuating and sealing three times. After cooling to room temperature, the resulting block copolymers were removed, dissolved in chloroform, and precipitated in excess of cold ethanol with mechanical stirring. The resulting PECL copolymers were dried at 38 °C under vacuum for 48 h. Cetyl alcohol initiated CL polymerization (CPCL) were synthesized with the same method above as a control. ¹H-NMR spectra (in CDCl₃) were recorded on a Bruker Fourier 300 apparatus at 25 °C to determine the copolymer molecular weight and the chemical composition. IR spectra were obtained on a Nicolet IR spectrometer to determine the chemical functional groups. The thermal properties of PECL and CPCL copolymers were measured on a differential scanning calorimeter (DSC, SEIKO) under a nitrogen atmosphere at a scanning rate of 10 °C/min from −80 to 80 °C. The contact angle of the vacuum-dried casting copolymers membrane was measured using a KRUSS DSA30 instrument.

**Fabrication of PECL microfibrous scaffold.** The direct writing device used to fabricate microfibers was custom-made by the Foshan Qingzinano Corporation primarily using a pneumatic extrusion system with a stainless syringe, a precise X–Y translational stage with a collector plate (conductive glass) to execute the pre-designed patterning route (e.g., straight, grid, curly, and arbitrary), and a high voltage power. A high-speed camera (HuaGuDongLi, SHL-200WS) was used to observe the morphology and motion of the jet. One gram of synthesized PECL copolymer was heated to 105 °C for 10 min and extruded at an air pressure of 20 kPa. With the 3 mm distance between the syringe needle (30G stainless) and the collector plate, and 3.6 kV electrostatic voltage, a stable PECL jet was obtained. The jet and fiber morphology under the stage velocity of 0, 20, 40, 60 and 80 mm/s were determined. The grid fibrous scaffolds were fabricated with a fiber spacing of 50, 100, 200, 300, 400, and 500 μm, and a scaffold height of 100 μm to mimic the orthogonally aligned stromal layer. The morphology of the writing fibers was imaged using a microscope with a super-wide depth of field (YEYENCE, VHX-6000) and Scanning Electron Microscope (SEM, Phenom XL) at 10 kV (Samples were pre-coated with a 20-nm-thick layer of platinum). The average fiber diameters were digitized and analyzed by Image Pro Plus (IPP) software. LSSCs inoculated on the grid fibrous scaffold were observed by cytoskeleton/nuclei staining.

**Preparation of GelMA hydrogel and fiber hydrogel construct.** GelMA was synthesized as previously described[56,57]. Briefly, 10 g type-A gelatin power (Sigma-Aldrich) from porcine skin was dissolved in 100 mL DPBS (Gibco) at 60 °C and stirred for 3 h before the addition of 6 mL methacrylic anhydride (MA, Sigma-Aldrich) at a rate of 0.1 mL/min and allowing the reaction to proceed for 4 h at 50 °C. The resulting mixture was diluted with 300 mL additional warm DPBS to stop the methacrylation reaction. After dialyzing against deionized water at 40 °C for 7 days using a 12–14 kDa cut-off dialysis membrane, the PH of the mixture solution was adjusted to 7.4 using 1 M NaHCO₃ and then the mixture solution was sterilized using 0.2 μm syringe filter and lyophilized for 72 h to obtain the GelMA sponge. GelMA hydrogel at 5, 10, and 15% (w/v) were prepared by dissolving the GelMA sponge in DPBS with 0.05% LAP at 60 °C and exposing to 365 nm light for 90 s. After encapsulating LSSCs within the 5%, 10%, and 15% (w/v) GelMA hydrogel for 3 days, respectively, Live/Dead assays and cytoskeleton/nuclei staining were performed to assess the cell viability and morphology within the GelMA hydrogel. The fiber hydrogel construct was fabricated by perfusing the GelMA solution into the direct writing grid microfibrous scaffold with a fiber spacing of 50, 100, 200, 300, 400, and 500 μm, respectively within a custom-made poly (methyl methacrylimide) mold with a 20-mm diameter and 500-μm height. The 50G, 100G, 200G, 300G, 400G, and 500G represented fiber hydrogels constructed with 50, 100, 200, 300, 400, and 500 μm grid microfibrous scaffolds, respectively, in the following content. The cross-section morphology of the GelMA hydrogel and 100G fiber hydrogel was imaged using the SEM at 15 kV (Samples were pre-coated with a 20-nm-thick layer of platinum) after freeze drying.

**Physiochemical characterization of fiber hydrogel.** The tensile and compressive properties of the six fiber hydrogels, acellular native cornea, and the GelMA hydrogel (only in the compressive test) were measured with a uniaxial load test machine (Electropuls E 3000, INSTRON) under the loading speeds of 5 mm/min and 0.05 mm/min for tensile and compressive tests, respectively. Tensile samples with rectangular shape (15 mm × 5 mm) and a thickness of 500 μm were prepared in a rectangular mold before immersing into PBS (PH = 7.4) for 1 h (day 0), 7, 14, and 28 days at 37 °C. Cylindrical compressive samples with a diameter of 8 mm and a thickness of 500 μm were prepared by using trephine to punch the fiber hydrogel formed in the 20 mm diameter mold before immersing into PBS (PH = 7.4) for 1 h (day 0). The samples were preconditioned by loading–unloading for 10 times under the same load level. The strain–stress tensile curves were plotted, and the maximum tensile strength and elongation at break, and the compressive modulus were recorded and calculated. Three samples were measured for each construct to calculate the mean and standard deviation.

The six fiber hydrogels and acellular cornea were prepared using a trephine to punch into the cylinder with a height of 500 μm and a diameter of 6.5 mm (n = 3) to allow to study in a 96-well plate. After immersion in PBS (PH = 7.4) at 37 °C for 0, 7, 14, and 28 days, the light transmittance of the fiber hydrogels and cornea were measured using a microplate reader (Multiskan Spectrum, Thermo Scientific) under wavelengths ranging between 400 and 750 nm.

The six fiber hydrogels with 500 μm height and 20 mm diameter and acellular cornea (n = 3) were prepared for swelling analysis. The samples were immersed in PBS (PH = 7.4) at 37 °C for 24 h to reach equilibrium swelling, and the swollen samples were weighed after gently wiping off the excess liquid with Kimwipes. The weighed samples were rapidly frozen in liquid nitrogen, lyophilized for 24 h, and reweighed. The swelling ratio was calculated by dividing wet weight by dry weight.

**Isolation, culture and inoculation of LSSCs.** Animal care, breeding, and euthanasia were performed with the approval of the Animal Ethics Committee of Tsinghua University. Cornea-scleral rims (limbus, ~2 mm width) were obtained from the eyeballs of freshly sacrificed adult SD rats. The rims were rinsed twice in PBS after cutting into 6–8 segments. The segments were incubated in 10 mg/mL dispase II neutral protease (Gibco) at 37 °C for 0.5 h and then rinsed twice in PBS. Segments were then immersed in 1 mg/mL collagenase (Gibco) at 37 °C for 1 h, rinsed twice with PBS, and then centrifuged at 4 °C. Supernatant was aspirated, and segments were incubated in trypsin (Sigma-Aldrich) for 15 min at 37 °C. Then, add the limbal stromal stem cell media (LSSCM, made of DMEM/F-12 (Gibco) supplemented with 10% knockout serum (Gibco), 1% ITS-X (Gibco), 4 ng/mL basic fibroblast growth factor (β-FGF, Peprotech), 10 ng/mL human leukemia inhibitory factor (LIF, Peprotech), 100 U/mL penicillin and 100 μg/mL streptomycin (Gibco)), samples

were centrifuged, supernatant was aspirated, and segments were re-suspended with LSSCM and seeded into collagen-coated twelve-well plates. Culture media were changed every 2–3 days, and cells were sub-cultured by digestion with trypsin at 90% confluence and plated into six-well plates, 25-cm$^2$ T-flasks, and 75-cm$^2$ T-flasks at P 1, P 2, and P 3, respectively. P 3 cells were used for experiments.

The obtained P 3 LSSCs were inoculated on the collagen-coated grid PECL microfibrous scaffold for one day in LSSCM before the perfusion of GelMA hydrogel to form a fiber hydrogel with cells. After culturing for one day, the LSSCM was replaced by serum-containing media (SCM, made of DMEM (Gibco) supplemented with 10% fetal bovine serum (Gibco), 100 U/mL penicillin, and 100 μg/mL streptomycin) and serum-free media (SFM, made of advanced DMEM/F12 with 1X GlutaMAX (Gibco), supplemented with 1 mM ascorbic acid (Sigma-Aldrich), 1% ITS-X (Gibco) and 10 ng/mL β-FGF), respectively.

**Immunostaining and Sirius staining**. LSSCs inoculated on 2D TCP, 3D GelMA, and 100G under SCM and SFM were observed to study the effects of the substrate topological structure and chemical factors on the differentiation of LSSCs, the maintenance of the keratocyte phenotype, and the secretion of keratocyte-specific extra cellular matrix.

After culturing for 1 week on 2D TCP, 3D GelMA, and 100G under SCM and SFM respectively, the samples were rinsed with PBS once and fixed with 4% paraformaldehyde (PFA) for 20 min. The samples were subsequently permeabilized in 0.1% Triton X-100 (Sigma-Aldrich) for 30 min and incubated with 1% (w/v) bovine serum albumin (Sigma-Aldrich) for 30 min to block non-specific binding. Samples in SCM and SFM were incubated overnight at 4 °C with the primary antibodies against Vimentin (1:100, Proteintech) and ALDH3A1 (1:50, Proteintech), respectively, and then incubated with the phalloidin-iFluor 555 (1:1000, Abcam) and secondary antibody goat anti-rabbit (Alexa Fluor 488, 1:1000, Abcam) for 1 h at room temperature. Lastly, the samples were stained with DAPI (Vectorlabs) before the observation by Olympus confocal microscope. Three replicates were used for each sample. After culturing for 4 weeks on 2D TCP, 3D GelMA, and 100G in SFM, the samples were processed as in the protocol detailed above except for the incubation with primary antibody against Collagen VI (1:200, Abcam) without the phalloidin staining.

Except for the immunostaining, after culturing LSSCs for 4 weeks on 2D TCP, 3D GelMA, and 100G in SFM, the samples were also stained with Sirius dye to further examine the secretion of collagen fibers. The staining process was performed according to the instructor's description. In brief, the samples were fixed in 4% PFA for 20 mins followed with the wash of PBS for three times. After the dehydration using 30% sucrose for 24 h at 4 °C, the samples were immersed into the Sirius dye solution for 1 h followed with the wash of water for few seconds. Then the samples were immersed into the hematoxylin solution for 10 mins and next washed with water for 10 mins. Next, 80%, 90%, and 100% ethanol and xylene were used to dehydrate and clear. Finally, the samples were observed using bright-field microscope.

**Gene expression**. After inoculating on 2D TCP, 3D GelMA, and 100G in SCM and SFM, respectively, for 2 weeks, cells were lysed by Trizol (Invitrogen) to extract RNA. The concentrations of RNA were measured using a Nanodrop 2000 (Thermo scientific). The same amount of RNA was transcribed to cDNA by using a Quantitative Reverse Transcription kit (Takara) according to the manufacturer's protocol. TaqMan gene expression master mix (Takara) and the primers for the studied gene (Supplementary, Table 1) were added to sample cDNA (2 μL), and a StepOnePlus real time PCR system (Applied Biosystems) instrument was used with 40 amplification cycles. Cells cultured on 2D TCP were used as a reference for comparison. β-actin was utilized as an endogenous control for the normalization of studied genes expression levels using the delta delta Ct (ΔΔCt) method. Calculation of $2^{-\Delta\Delta Ct}$ was performed to give a comparative fold change of gene expression level relative to actin. Three replicates were measured for each sample to calculate the mean and standard deviation.

**Rat intrastromal keratoplasty and evaluation**. The animals were maintained in a temperature-controlled environment (20 ± 1 °C) with free access to food and water. Animal care, breeding, and euthanasia were performed with the approval of the Animal Ethics Committee of Tsinghua University.

Thirty, ~4-weeks-old male SD rats weighing 0.1–0.2 kg, which were divided into five groups of six each, and intrastromal keratoplasty was performed. The rats were anesthetized by the abdominal injection of 10% chloralic hydras (Leagene). Local anesthetic drops (Tetracaine hydrochloride) were also used during the surgery. The left eyes underwent intra-stromal corneal transplantation in the thirty rats, and right eyes served as uninjured controls. In all groups, surgical scissors were used to create an aperture along the corneal circumference, and a tunnel knife was used to excise a layer of stroma and leave an empty stromal pocket. 100G (with and without chemical factors, which included β-FGF, ascorbic acid and insulin, with the same concentration as in the SF media) with a diameter of 1.5 mm and height of 100 μm was squeezed into the pocket before suturing the aperture in group 1 and group 2, respectively. 3D GelMA hydrogel (with and without chemical factors) was transplanted using the same method in group 3 and group 4. In group 5, the excised native stromal layer was inserted into its original position (autograft

transplantation) as a positive control. After the surgery, erythromycin ointment and dexamethasone eye drops were applied to the surgical eye three times per day for the first week, and usage was gradually reduced in the following three months. The first week after operation, all rats were visually observed daily for clinical signs of inflammation and the growth of blood vessels. Detailed in vivo examinations under general anesthesia were performed at 1, 4, 8, and 12 weeks after operation to visualize the grafts, neovascularization, and light transmittance using a surgical operating microscope.

Immediately, 1, 4 and 12 weeks after transplantation, the operated cornea were photographed with a slit lamp (YZ5S, 66 Vision Tech, China). A self-assembly Optical Coherence Tomography was used to evaluate the intra-stromal location of the transplanted construct and the thickness of the cornea (three rats in each condition) immediately, 4 and 12 weeks after operation. In all, 4 and 12 weeks after operation, three rats for each time and each group were sacrificed, and the implanted regions were excised and fixed in 4% PFA to prepare frozen sections for immunostaining and paraffin sections for hematoxylin and eosin (H&E) staining. In brief, for immunostaining, after fixation in PFA for 12 h, the samples were dehydrated by immersing into 30% sucrose for 24 h at 4 °C. Sample were filled with OCT compound, placed on the surface of liquid nitrogen to freeze, and sectioned to 10 μm thickness using a cryostat. The sections were blocked with 1% BSA and incubated overnight at 4 °C with the primary antibodies against collagen VI and then incubated with the secondary antibody goat anti-rabbit (Alexa Fluor 488) for 1 h at room temperature. Lastly, the samples were coverslipped with mounting medium containing with DAPI before the observation by Olympus confocal microscope under ×10 and ×20 magnification. For H&E staining, the deparaffinized sections were cleared in xylene and rehydrated in ethanol. Rinse briefly in distilled water before the staining in hematoxylin solution. Next, rinse the sections with running tap water followed by differentiating in 1% acid alcohol and washing with running tap water. Bluing in 1% ammonia water before rinsing with running tap water. Then, samples were stained in eosin solution before the dehydration with ethanol and clearing with xylene. The samples finally were observed by Olympus microscope.

**Reporting summary**. Further information on research design is available in the Nature Research Reporting Summary linked to this article.

## Data availability
The data that support the findings of this study are available from the corresponding author upon reasonable request. Raw data of the following figures is attached in the supplementary source data: Figs. 2b–f, 3c–h, j, and Fig. 4c, Supplementary Figs. 5 and 6.

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

## Acknowledgements

We acknowledge financial support from the National Key Research and Development Program of China (No. 2016YFC1100100& 2018YFC1106000), the Key Research and Development Projects of People's Liberation Army (No. BWS17J036), the Project of Basic Research of Shenzhen, China (JCYJ20170412101508433 & JCYJ20180507183655307) and Key Research and Development Plan Project of Shaanxi province (2018ZDXM-SF-056).

## Author contributions

B.K., W.S., and S.M. outlined the paper, conceived, and designed the experiments. B.K. wrote the main paper text, did most experiments. R.L. accomplished the design of the direct writing device and the fabrication of the fibers. Y.C. and X.L. performed the animal transplantation. C.L. helped to design the directing writing device. Z.S., L.X., and X.L. supplied the method to isolate the cells. All authors discussed the results and commented on the paper.

## Competing interests

The authors declare no competing interests.
