## [Peer Review File · Nature Communications]

Reviewers' Comments:

Reviewer #1:

Remarks to the Author:

The manuscript should undergo major revision based on the following comments and additional experiments analysis to be performed.

The analysis of the in vivo data set needs further investigation as it lacks any quantitative data sets.

Mechanical analysis and results should be discussed to the state of the art in the field , see following papers

An Integrated Design, Material, and Fabrication Platform for Engineering Biomechanically and Biologically Functional Soft Tissues.

Bas O, D'Angella D, Baldwin JG, Castro NJ, Wunner FM, Saidy NT, Kollmannsberger S, Reali A, Rank E, De-Juan-Pardo EM, Hutmacher DW.

ACS Appl Mater Interfaces. 2017 Sep 6;9(35):29430-29437. doi: 10.1021/acsami.7b08617. Epub 2017 Aug 23.

PMID:28816441

Biofabricated soft network composites for cartilage tissue engineering.

Bas O, De-Juan-Pardo EM, Meinert C, D'Angella D, Baldwin JG, Bray LJ, Wellard RM, Kollmannsberger S, Rank E, Werner C, Klein TJ, Catelas I, Hutmacher DW.

Biofabrication. 2017 May 12;9(2):025014. doi: 10.1088/1758-5090/aa6b15.

PMID:28374682

The reference to GELMA preparation is inadequate

144. reparation of GelMA hydrogel and fiber hydrogel construct

145. 145 GelMA was synthesized as previously described²⁰.

Reference 20 Ouyang LL, Highley CB, Sun W, Burdick JA. A Generalizable Strategy for the 3D Bioprinting of Hydrogels from Nonviscous Photo-crosslinkable Inks. *Advanced materials* 29, (2017).

Describes only briefly the synthesis of GELMA. This reference

Functionalization, preparation and use of cell-laden gelatin methacryloyl-based hydrogels as modular tissue culture platforms.

Loessner D, Meinert C, Kaemmerer E, Martine LC, Yue K, Levett PA, Klein TJ, Melchels FP, Khademosseini A, Hutmacher DW.

Nat Protoc. 2016 Apr;11(4):727-46. doi: 10.1038/nprot.2016.037. Epub 2016 Mar 17.

PMID: 26985572

gives a detailed protocol

It should be added in the discussion that all biomaterials used were not medical grade and hence could not be used clinically contrary to what is stated in manuscript that this concept can be translated to the clinic.

The appropriateness to underline the argument of this references

Mi SL, Kong B, Wu ZJ, Sun W, Xu YY, Su X. A novel electrospinning setup for the fabrication of thickness-controllable 3D scaffolds with an ordered nanofibrous structure. *Mater Lett* 160, 343-346 (2015).

Sun DH, Chang C, Li S, Lin LW. Near-field electrospinning. *Nano Lett* 6, 839-842 (2006).

Hochleitner G, et al. Additive manufacturing of scaffolds with sub-micron filaments via melt electrospinning writing. *Biofabrication* 7, (2015).

Chen F, Hochleitner G, Woodfield T, Groll J, Dalton PD, Amsden BG. Additive Manufacturing of a Photo-Cross-Linkable Polymer via Direct Melt Electrospinning Writing for Producing High Strength Structures. *Biomacromolecules* 17, 208-214 (2016).

should be considered again

Figure 1 & 2 should be moved to supplementary information.

This review The quest for mechanically and biologically functional soft biomaterials via soft network composites.

Bas O, Catelas I, De-Juan-Pardo EM, Hutmacher DW.

Adv Drug Deliv Rev. 2018 Jul;132:214-234. doi: 10.1016/j.addr.2018.07.015. Epub 2018 Jul 24. Review.

PMID:30048654

Summarizes the state of the art on the topic of this study.

The manuscript must be proof read by a native speaker.

Reviewer #2:

Remarks to the Author:

Recommendation: Publish after major revisions noted.

Comments:

The authors fabricated the hydrogel with orthogonally aligned fibers and the serum-free media supplemented with the chemical factors which then was implanted into rat corneas for the maintenance of the keratocyte phenotype and the regeneration of the damaged corneal stroma. They showed that the fiber hydrogel can regulate the differentiation of LSSCs to keratocytes, maintain the keratocyte phenotype, and promote the secretion of keratocyte-specific ECM in vitro. Animal experiments showed that the combination of these factors can ultimately allow the regeneration of injured stromal tissue in vivo in 28 days. Although the work is quite interesting, and biomimetic corneal stroma was also obtained, the present form of this work cannot be accepted for the publication on Nature Communication. There some issues should be addressed before publication.

Major comments:

1. Animal experiment design is too simple. Thus, rabbit corneal transplant model within the longer transplantation period is recommended, which can ensure that the material has a chance to be used clinically.
2. The superiority for the combination of orthogonally aligned poly (ϵ -caprolactone)-poly (ethylene glycol) (PECL sub-micro fibers and GelMA hydrogel should be stated in introduction.
3. The authors mentioned that the collagen secreted on 2D was random, demonstrating no alignments, however, the collagen secreted on 100G exhibits orthogonal alignment along the grid fibers in line 459, but in Figure 8, it cannot be sure that the green orthogonal alignment represents the collagen VI secreted by LSSCs. Maybe it just can be dyed as the green. In addition, What does the red arrow represent?
4. In Fig. 9, the images by surgical microscope are meaningless, because we can't locate the material and the wound. The authors claimed that the synergistic effects of orthogonally aligned topological cues and specific chemical factors have been demonstrated, but it is uncertain whether they have the synergistic effects. It may be the effect of topological structure or chemical factors.

There are some minor issues

1. The author should delete "3 times" in line 115.
2. The author should delete "were determined" in line 136.
3. The authors should check that "weighted" is correctly written in line 186.
4. There are some grammatical errors in line 237-239.

5. The authors did not indicate the sex of the SD rats, and there are some grammatical errors in line 259.
6. The authors did not explain what "50G, 100G et. al" means in the article.
7. The authors mentioned Figure 10 in line 457, but we don't find Figure 10. Please explain the problem.
8. There too many spaces should be added between number and units.

Dear Reviewer #1:

Thank you very much for your helpful comments. I have already revised and improved the manuscript according to your suggestions. The following is the responds to your specific comments.

Response to the specific comments:

The manuscript should undergo major revision based on the following comments and additional experiments analysis to be performed.

1. The analysis of the in vivo data set needs further investigation as it lacks any quantitative data sets.

Response: thank you for your suggestion. We re-performed the rat corneal transplantation using 30 rats, which were divided into 5 groups of 6 each. The autograft, 3 D GelMA hydrogel, 3 D GelMA hydrogel with factors, 100 G fiber hydrogel and 100 G fiber hydrogel with factors were transplanted into the corneal stroma of the rats using intrastromal keratoplasty in the 5 groups, respectively, to systematically study the effect of the chemical factors and the topological fibers on the regeneration of stroma in vivo. In order to evaluate the intra-stromal location of the transplanted grafts and quantitatively characterize the change of the corneal thickness, a self-assembly Optical Coherence Tomography was used immediately, 4 and 12 weeks after operation, 3 rats were used in each situation to measure the corneal thickness. The new in vivo data is shown in the figure below.

Fig. 4 Fiber hydrogel engraftment and stromal matrix synthesis and regeneration in rat cornea in vivo. A) Images taken by slit lamp after operation, 1 and 3 months transplantation of the autograft, 3 D GelMA hydrogel with or without chemical factors, 100 G fiber hydrogel with and without factors. The white arrows indicate the position of the grafts. B) The OCT images of the same corneas in A) postoperatively. The white arrows indicate the position of the grafts. The scale bar is 1000 µm. C) The thickness of the central corneas by measuring three corneas in each group and each time point. D) On 1 month and 3

months, frozen sections of the transplanted corneas were immunostained with collagen type VI antibodies (green). In all images, nuclei are co-stained with DAPI (blue), the scale bar is 100 μ m. E) H&E staining of the transplanted corneas on 1 and 3 months, the scale bar is 100 μ m.

The autograft, 3 D GelMA hydrogel, 3 D GelMA hydrogel with chemical factor, 100 G fiber hydrogel and 100 G fiber hydrogel with chemical factors were transplanted into corneas of 30 rats using the intrastromal keratoplasty. The white arrows in Figure 4A indicated the positions of these transplanted grafts. One week after operation, severe inflammation was observed in a rat with 3 D hydrogel, and a rat with 100 G fiber hydrogel. They were pre-sacrificed because the symptom didn't subside or get better after treatment with antibiotics. Besides, 4 rats died accidentally 1 month and 2 months after surgical operation, and we re-transplanted the grafts. 1 month postoperatively, 4 of 30 corneas were observed with neovascularization (two in 3 D hydrogel, one in 3 D with factors and one in 100 G fiber hydrogel), and the symptom disappeared after 3 months without any treatment. From the images taken by slit lamp shown in Figure 4A, 3 months postoperatively, the transplanted corneas all showed high light transmittance. From the OCT images shown in Figure 4 B, we can clearly observe the position of the transplanted grafts (white arrow) and the dots in the 100 G with or without factors indicated the PECL fibers. Immediately after operation, all corneas showed apparent edema and significant increase of the thickness (Figure 4 C). 1 month after operation, the edema subsided and the thickness of all corneas have decreased, with anterior curvature following that of the posterior cornea. 3 months after operation, the thickness of all corneas continually decreased but still had a little larger than that of the native rats. With the increase of time, the area of the blank occupied by the grafts all decreased, indicating that the degradation of the scaffolds and the regeneration of the stroma.

1 month after operation, histological H&E staining and collagen VI immunofluorescence staining (Figure 4 D and E) both revealed that the graft with factors can induce the regeneration of the stroma better than the same graft without factors. And the graft with fibers can induce the regeneration of the stroma better than the same graft without fibers. The white arrows in 100 G and 100 G with factors indicated the PECL fibers, because the fibers cannot be fully degraded within 3 months. However, when comparing the 100 G and 100 G with factors, we can find out that the occupied positions by fibers in 100 G with factors was smaller than that in the 100 G, also indicating that the factors may accelerate the degradation of the fibers by stimulating the regenerated keratocytes to secrete the degrading enzyme. 3 months after operation, the influence of the factors and fibers on the regeneration of the stroma was same with that in 1 month, and in the 3 D and 3 D with factors, the stroma were not fully regenerated. However, 100 G and 100 G with factors have fully induced the regeneration of the stromal tissue with normal morphology except for the positions occupied by the undegraded fibers. All results indicated that the synergistic effects of 100 G and chemical factors can provide the most suitable topographical and chemical environment to induce the regeneration of the corneal stroma *in vivo*.

2. Mechanical analysis and results should be discussed to the state of the art in the field, see following papers

An Integrated Design, Material, and Fabrication Platform for Engineering Biomechanically and Biologically Functional Soft Tissues. Bas O, D'Angella D, Baldwin JG, Castro NJ, Wunner FM, Saily NT, Kollmannsberger S, Reali A, Rank E, De-Juan-Pardo EM, Huttmacher DW. ACS Appl Mater Interfaces. 2017 Sep 6;9(35):29430-29437. doi: 10.1021/acsami.7b08617. Epub 2017 Aug 23. PMID:28816441

Biofabricated soft network composites for cartilage tissue engineering. Bas O, De-Juan-Pardo EM, Meinert C, D'Angella D, Baldwin JG, Bray LJ, Wellard RM, Kollmannsberger S, Rank E, Werner C, Klein TJ, Catelas I, Huttmacher DW. Biofabrication. 2017 May 12;9(2):025014. doi: 10.1088/1758-5090/aa6b15. PMID:28374682

Response: thank you for your suggestion. We are regretful that we missed so important papers closely related to our study when investigating literatures. We read these papers carefully and discussed the mechanical results according to the art state in the field. You can see the newly added discussion with the yellow background in the content below.

The tensile and compressive strength, the light transmittance, and the swelling ratio of the 50 G, 100 G, 200 G, 300 G, 400 G, and 500 G constructs were measured to study the effects of fiber spacings on these properties and to verify the optimal spacing for producing a construct with properties most similar to the native corneal tissue. The tensile strain-stress curves and elongation at break of fiber hydrogels and acellular porcine cornea after immersing into PBS for 1 h (day 0) are shown in Figure 2 A and B; a large elongation at near constant stress were observed from the strain-stress curves of the fiber hydrogels after the materials yielded, which is similar to that of a typical flexible plastic⁴⁴; and with increased fiber spacing, the maximum tensile stress and elongation at break both decreased. The 50 G and 100 G constructs can attain the tensile strength requirements of the native cornea (3-5 MPa)⁴⁵, with the maximum tensile strengths of 3.79 ± 0.51 and 3.47 ± 0.43 MPa, respectively. All constructs had high elongation at break, indicating that the fiber hydrogels had high ductility much greater than that of the native cornea. The maximum tensile strength of fiber hydrogels after immersing into PBS for 7, 14, and 28 days were also measured to verify the long-term mechanical stability of the constructs, as shown in Figure 2 C. With increased immersion time, the tensile strength of all constructs decreased to a certain degree. However, the 50 G and 100 G constructs can still meet the requirements of the native cornea with the tensile strengths of 3.42 ± 0.46 and 2.93 ± 0.36 MPa after 28 days. Figure 2 D shows the compressive modulus of pure GelMA hydrogel, fiber hydrogels constructs, and the native porcine cornea. The addition of grid fibers can dramatically enhance the compressive modulus of GelMA, and the smaller the fiber spacing, the larger the modulus. The effect trend is consistent with previous research results^{35,36}. Although the fibrous scaffolds alone can bear limited compressive strength due to the intrinsic properties, including fiber delamination and buckling, for the fiber hydrogels, the compression would cause the stretch of hydrogels within the grid fibrous scaffold, thus, the longitudinal compressive force would transfer to transverse tensile force to the fibrous scaffold. Due to the excellent tensile strength of the fibrous scaffolds, the fiber hydrogels can also have better compressive strength³⁶. Except for 400 G

and 500 G, other fiber reinforced constructs all have the compressive modulus similar to or greater than the native corneal tissue.

Thus, we designed fibrous grid scaffolds with six different fiber spacings, ranging from 50 to 500 μm to study the relationship between the fiber spacing and construct properties and to identify the optimal fiber spacing. From the results shown in Figures 2 and Supplementary Figure 3, we easily discerned that the maximum tensile stress, elongation at break, and compressive modulus of the constructs were inversely proportional to the fiber spacing, and the light transmittance and mass swelling ratio were proportional to the fiber spacing. These phenomena can be consistently interpreted as fiber numbers effects. When the shape and area of samples were the same for one measurement, the smaller fiber spacing means the more fibers existing in the construct. With more fibers, the construct will undoubtedly have better mechanical properties because the fibrous scaffold supplies most of the mechanical force in the construct. It should be recognized that by reinforcing with orthogonally aligned fibers, we can significantly improve the maximum tensile strength of the fiber hydrogels and attain the strength requirement of the native corneal tissue. But from Figure 2A and previous research results⁴⁴, we can find out that the native soft tissues exhibit a J-shaped strain-stress curves with three distinct phases, which is different from that of the designed fiber hydrogels. The difference may because the straight fibers were already taut, and the stress would rapidly increase without a gradual increase when the stretch began. Onur Bas et al used near-field electrospinning to fabricate curved fibers, and have demonstrated that the reinforced hydrogels with curved fibers successfully displayed a J-shaped strain-stress curves⁴⁴. And future effort can be applied to the design of the fiber patterns to make a construct with mechanical properties most similar to the native corneal tissues.

3. The reference to GELMA preparation is inadequate

144. preparation of GelMA hydrogel and fiber hydrogel construct

145. 145 GelMA was synthesized as previously described²⁰.

Reference 20 Ouyang LL, Highley CB, Sun W, Burdick JA. A Generalizable Strategy for the 3D Bioprinting of Hydrogels from Nonviscous Photo-crosslinkable Inks. *Advanced materials* 29, (2017).

Describes only briefly the synthesis of GELMA. This reference

Functionalization, preparation and use of cell-laden gelatin methacryloyl-based hydrogels as modular tissue culture platforms. Loessner D, Meinert C, Kaemmerer E, Martine LC, Yue K, Levett PA, Klein TJ, Melchels FP, Khademhosseini A, Huttmacher DW. *Nat Protoc.* 2016 Apr;11(4):727-46. doi: 10.1038/nprot.2016.037. Epub 2016 Mar 17. PMID: 26985572

gives a detailed protocol

Response: thank you for your suggestion. We have read the paper carefully and cited in the revised manuscript.

4. It should be added in the discussion that all biomaterials used were not medical grade and hence could not be used clinically contrary to what is stated in manuscript that this concept can be translated to the clinic.

Response: thank you for your careful consideration. We have added that all biomaterials used were not medical grade and hence could not be used clinically in the discussion part of the revised manuscript.

5. The appropriateness to underline the argument of this references

Mi SL, Kong B, Wu ZJ, Sun W, Xu YY, Su X. A novel electrospinning setup for the fabrication of thickness-controllable 3D scaffolds with an ordered nanofibrous structure. *Mater Lett* 160, 343-346 (2015).

Sun DH, Chang C, Li S, Lin LW. Near-field electrospinning. *Nano Lett* 6, 839-842 (2006). Hochleitner G, et al. Additive manufacturing of scaffolds with sub-micron filaments via melt electrospinning writing. *Biofabrication* 7, (2015).

Chen F, Hochleitner G, Woodfield T, Groll J, Dalton PD, Amsden BG. Additive Manufacturing of a Photo-Cross-Linkable Polymer via Direct Melt Electrospinning Writing for Producing High Strength Structures. *Biomacromolecules* 17, 208-214 (2016).

should be considered again

Response: thank you for your suggestion. We read these papers again and found out that it is indeed not appropriate to underline the argument of these papers. We removed these papers.

6. Figure 1 & 2 should be moved to supplementary information.

Response: thank you for your suggestion. We removed Figure 1 to the supplementary material. Figure 2 shows the characterization of the PECL direct writing fibrous scaffold, which is one of the key novelty of our study, thus, we still put it in the main manuscript and merged with Figure 3 to reduce the total number of the figures.

7. This review The quest for mechanically and biologically functional soft biomaterials via soft network composites. Bas O, Catelas I, De-Juan-Pardo EM, Hutmacher DW. *Adv Drug Deliv Rev.* 2018 Jul;132:214-234. doi: 10.1016/j.addr.2018.07.015. Epub 2018 Jul 24. Review. PMID:30048654

Summarizes the state of the art on the topic of this study.

Response: thank you for your suggestion. We feel sorry that we missed so important review closely related to our study when investigating literatures. We read this paper carefully and learned a lot, and also cited this paper in the revised manuscript.

8. The manuscript must be proof read by a native speaker.

Response: thank you for your suggestion. We have invited a native speaker to help us to revise the language.

Dear Reviewer #2:

Thank you very much for your careful and detailed review. You have given many useful suggestions which can greatly improve the quality of this manuscript. We have made elaborate and complete revisions or improvements according to your specific constructive comments.

Response to the specific comments:

Recommendation: Publish after major revisions noted.

Comments:

The authors fabricated the hydrogel with orthogonally aligned fibers and the serum-free media supplemented with the chemical factors which then was implanted into rat corneas for the maintenance of the keratocyte phenotype and the regeneration of the damaged corneal stroma. They showed that the fiber hydrogel can regulate the differentiation of LSSCs to keratocytes, maintain the keratocyte phenotype, and promote the secretion of keratocyte-specific ECM in vitro. Animal experiments showed that the combination of these factors can ultimately allow the regeneration of injured stromal tissue in vivo in 28 days. Although the work is quite interesting, and biomimetic corneal stroma was also obtained, the present form of this work cannot be accepted for the publication on Nature Communication. There some issues should be addressed before publication.

Major comments:

1. Animal experiment design is too simple. Thus, rabbit corneal transplant model within the longer transplantation period is recommended, which can ensure that the material has a chance to be used clinically.

Response: thank you for your suggestion. We don't have enough space to raise many rabbits, so we continually used rat to perform the corneal transplantation. But, in the newly designed experiment, the autograft, 3 D GelMA hydrogel, 3 D GelMA hydrogel with factors, 100 G fiber hydrogel and 100 G fiber hydrogel with factors were transplanted into the corneal stroma of the rats using intrastromal keratoplasty for 3 months, respectively, to systematically study the effect of the chemical factors and the topological fibers on the regeneration of stroma in vivo. 30 rats were used and each group had 6 rats. The new in vivo data is shown in the figure below.

Fig. 4 Fiber hydrogel engraftment and stromal matrix synthesis and regeneration in rat cornea in vivo. A) Images taken by slit lamp after operation, 1 and 3 months transplantation of the autograft, 3 D GelMA hydrogel with or without chemical factors, 100 G fiber hydrogel with and without factors. The white arrows indicate the position of the grafts. B) The OCT images of the same corneas in A) postoperatively. The white arrows indicate the position of the grafts. The scale bar is 1000 μm . C) The thickness of the central corneas by measuring three corneas in each group and each time point. D) On 1 month and 3 months, frozen sections of the transplanted corneas were immunostained with collagen type VI antibodies (green). In all images, nuclei are co-stained with DAPI (blue), the scale bar is 100 μm . E) H&E staining of the transplanted corneas on 1 and 3 months, the scale bar is 100 μm .

The autograft, 3 D GelMA hydrogel, 3 D GelMA hydrogel with chemical factor, 100 G fiber hydrogel and 100 G fiber hydrogel with chemical factors were transplanted into corneas of 30 rats using the intrastromal keratoplasty. The white arrows in Figure 4A indicated the positions of these transplanted grafts. One week after operation, severe inflammation was observed in a rat with 3 D hydrogel, and a rat with 100 G fiber hydrogel. They were pre-sacrificed because the symptom didn't subside or get better after treatment with antibiotics. Besides, 4 rats died accidentally 1 month and 2 months after surgical operation, and we re-transplanted the grafts. 1 month postoperatively, 4 of 30 corneas were observed with neovascularization (two in 3 D hydrogel, one in 3 D with factors and one in 100 G fiber hydrogel), and the symptom disappeared after 3 months without any treatment. From the images taken by slit lamp shown in Figure 4A, 3 months postoperatively, the transplanted corneas all showed high light transmittance. From the OCT images shown in Figure 4 B, we can clearly observe the position of the transplanted grafts (white arrow) and the dots in the 100 G with or without factors indicated the PECL fibers. Immediately after operation, all corneas showed apparent edema and significant increase of the thickness (Figure 4 C). 1 month after operation, the edema subsided and the thickness of all corneas have decreased, with anterior curvature following that of the posterior cornea. 3 months after

operation, the thickness of all corneas continually decreased but still had a little larger than that of the native rats. With the increase of time, the area of the blank occupied by the grafts all decreased, indicating that the degradation of the scaffolds and the regeneration of the stroma.

1 month after operation, histological H&E staining and collagen VI immunofluorescence staining (Figure 4 D and E) both revealed that the graft with factors can induce the regeneration of the stroma better than the same graft without factors. And the graft with fibers can induce the regeneration of the stroma better than the same graft without fibers. The white arrows in 100 G and 100 G with factors indicated the PECL fibers, because the fibers cannot be fully degraded within 3 months. However, when comparing the 100 G and 100 G with factors, we can find out that the occupied positions by fibers in 100 G with factors was smaller than that in the 100 G, also indicating that the factors may accelerate the degradation of the fibers by stimulating the regenerated keratocytes to secrete the degrading enzyme. 3 months after operation, the influence of the factors and fibers on the regeneration of the stroma was same with that in 1 month, and in the 3 D and 3 D with factors, the stroma were not fully regenerated. However, 100 G and 100 G with factors have fully induced the regeneration of the stromal tissue with normal morphology except for the positions occupied by the undegraded fibers. All results indicated that the synergistic effects of 100 G and chemical factors can provide the most suitable topographical and chemical environment to induce the regeneration of the corneal stroma in vivo.

2. The superiority for the combination of orthogonally aligned poly (ϵ -caprolactone)-poly (ethylene glycol) (PECL sub-micro fibers and GelMA hydrogel should be stated in introduction.

Response: thank you for your suggestion. We have added the superiority of the fiber hydrogel in the introduction.

Hydrogels from natural materials are also widely used in the construction of corneal tissue due to their excellent biocompatibility, intrinsic cell binding sites, 3 D highly porous structure and highly aqueous environment. However, the poor mechanical stability of natural hydrogels can limit their application in tissue engineering. Hydrogels from synthetic materials have better mechanical strength, but worse biocompatibility. Thus, to obtain a hydrogel with good biocompatibility and mechanical properties simultaneously, semi-synthetic and chemically-functionalized hydrogels (e.g. gelatin methacrylate (GelMA)) emerged. Recently, the combination of hydrogels and fibrous scaffold has been used to engineer fiber reinforced hydrogels applied in the regeneration of soft tissues, including cartilage, heart valve, tendon and muscle. The fibrous structure of the fiber hydrogel construct can resemble and simulate the biological fibers of the native soft tissues and supply the hierarchical and morphological cues to induce the specific growth and differentiation of cells. Besides, the fibrous scaffolds can enhance the mechanical performance of the hybrid construct by transferring the most load from the hydrogels to the fibers, and the mechanical strength can be easily regulated by changing the materials and processing parameters to engineer a construct with structure and mechanical properties most similar to the

targeted soft tissues, especially for the fibrous scaffolds fabricated from the near field electrospinning. The hydrogel of the fiber hydrogel construct can resemble the GAG or PG matrix and supply a real 3 D aqueous and porous environment for the cells to adhere, migrate, proliferate and differentiate. And the semi-synthetic hydrogels can also supply a highly regulatory and controllable chemical modifications for specific cells and tissues.

3. The authors mentioned that the collagen secreted on 2D was random, demonstrating no alignments, however, the collagen secreted on 100G exhibits orthogonal alignment along the grid fibers in line 459, but in Figure 8, it cannot be sure that the green orthogonal alignment represents the collagen VI secreted by LSSCs. Maybe it just can be dyed as the green. In addition, what does the red arrow represent?

Response: thank you for your suggestion. As to our knowledge, in the excitation of ultraviolet light, the grid PECL microfibrinous scaffold would exhibit blue in the confocal microscope, even without the staining of the dyes. Thus, you can see that the color of fibers was very dark in the DAPI staining images, this may result from the specific interaction of the grid structure of the microfibers with the ultraviolet light. However, in the excitation of the blue light, the grid microfibrinous scaffold wouldn't exhibit green in the confocal microscope without the staining of the dyes. It can also be dyed as green after staining due to the absorbance of the protein, but after washed the scaffold with excess PBS for 4 x 30min, the green color of the fibers would become very weak, thus, we used the red arrow to indicate the direction of the fibers and the green color shown in the image was the collagen VI secreted by the cells. And from the direction of most cell nucleus, we can find out most cells grow along the direction of the fibers, and thus the secreted collagen also exhibited orthogonal alignment along the grid fibers.

4. In Fig. 9, the images by surgical microscope are meaningless, because we can't locate the material and the wound. The authors claimed that the synergistic effects of orthogonally aligned topological cues and specific chemical factors have been demonstrated, but it is uncertain whether they have the synergistic effects. It may be the effect of topological structure or chemical factors.

Response: thank you for your suggestion. In the newly designed in vivo experiments, we used slit lamp to take the images of the cornea and we can observe the transparency, inflammation and neovascularization of the transplanted cornea through these images. Besides, a self-assembly OCT was used to locate the materials and wound. The autograft, 3 D GelMA hydrogel, 3 D GelMA hydrogel with factors, 100 G fiber hydrogel and 100 G fiber hydrogel with factors were transplanted into the corneal stroma of the rats using intrastromal keratoplasty for 3 months, respectively, to systematically study the effect of the chemical factors and the topological fibers on the regeneration of stroma in vivo. From the results shown in Figure 4, 1 month or 3 months after operation, histological H&E staining and collagen VI immunofluorescence staining both revealed that the graft with factors can induce the regeneration of the stroma better than the same graft without factors. And the graft with fibers can induce the regeneration of the stroma better than the same graft without fibers. The regeneration of the

stromal tissue was the best in the 100 G fiber hydrogel compared with other synthesized grafts, indicating that the synergistic effects of 100 G and chemical factors can provide the most suitable topographical and chemical environment to induce the regeneration of the corneal stroma in vivo.

5. There are some minor issues

- 1) The author should delete “3 times” in line 115.
- 2) The author should delete “were determined” in line 136.
- 3) The authors should check that “weighted” is correctly written in line186.
- 4) There are some grammatical errors in line 237-239.
- 5) The authors did not indicate the sex of the SD rats, and there are some grammatical errors in line 259.
- 6) The authors did not explain what “50G, 100G et. al” means in the article.
- 7) The authors mentioned Figure 10 in line 457, but we don’t find Figure 10. Please explain the problem.
- 8) There too many spaces should be added between number and units.

Response: thank you for your careful review.

We have corrected the errors in the revised manuscripts.

5) All the SD rats were male, we indicated in the materials and methods <Rat intrastromal keratoplasty and evaluation>.

6) We added the explanation in the results <Preparation of GelMA and fiber hydrogel construct>, The fiber hydrogel was fabricated by infusing the GelMA solution into the mold with the 50-500 um grid scaffold and named 50 G, 100 G, 200 G, 300 G, 400 G and 500 G, respectively.

7) It should be Figure 8, we wrote the wrong number.

Reviewers' Comments:

Reviewer #1:

Remarks to the Author:

The authors addressed most of the questions of the reviewers yet further questions need to be addressed in a second revision of the manuscript.

The following comments should be addressed.

In the intro and discussion, the authors cite and discuss several papers which use solution electrospinning to create fibre networks. It is a common failure in the literature that solution electrospun meshes are actually not made of nanofibers. Nanotechnology is science, engineering, and technology conducted at the nanoscale, which is about 1 to 100 nanometres. Physicist Richard Feynman, the father of nanotechnology.

"a large elongation at near constant stress were observed from the strain-stress curves of the fibre hydrogels after the materials yielded, which is similar to that of a typical flexible plastic44 "

Typical flexible plastic is not a scientific description. The authors should be more specific in which polymer is comparable!

"The collagen secreted on 2 D was random, demonstrating no alignments, however, the collagen secreted on 100 G exhibited orthogonal alignment along the grid fibers (the red lines in Figure 3I showed the504 direction of the fibers). These results indicated that the keratocytes can maintain their phenotype in SF media after culturing for 4 weeks, and the grid scaffold can induce the secretion of orthogonally aligned keratocyte specific ECM."

This is a statement is not supported by a quantitative data set. Yet it is a very important aspect in the regeneration of corneal tissues. The authors have to measure this alignment e.g. via picrosirius red staining of the collagen VI or second harmonic study protocols.

"The optimal fiber spacing of 100 um was verified to create a fiber hydrogel construct with the mechanical properties, light transmittance, and swelling ratio most similar to the native corneal tissue." It should be clarified that this results are for rat cornea. Human cornea might need other parameters to be optimal" E.g. sentence could be "The optimal fiber spacing of 100 um was verified to create a fiber hydrogel construct with the mechanical properties, light transmittance, and swelling ratio most similar to the native corneal tissue of rats. Future studies need to verify of this parameters would be the same for human applications."

Reviewer #2:

None

Dear Reviewer #1:

Thank you very much for your further suggestions and comments to our manuscript. We have already revised and improved the manuscript according to your suggestions. The following is the response to your specific comments.

Response to the specific comments:

Comments:

The authors addressed most of the questions of the reviewers yet further questions need to be addressed in a second revision of the manuscript.

Specific comments:

1. In the intro and discussion, the authors cite and discuss several papers which use solution electrospinning to create fibre networks. It is a common failure in the literature that solution electrospun meshes are actually not made of nanofibers. Nanotechnology is science, engineering, and technology conducted at the nanoscale, which is about 1 to 100 nanometres. Physicist Richard Feynman, the father of nanotechnology.

Response: thank you for your careful review. We feel regret that we didn't realize the true definition of nanotechnology and discussed the previous research results about solution electrospinning as nanofibers. We have revised the description about the related nanofibers to sub-microfibers or fibers and you can find out in the revised manuscript marked with yellow background.

2. "a large elongation at near constant stress were observed from the strain-stress curves of the fibre hydrogels after the materials yielded, which is similar to that of a typical flexible plastic44 "

Typical flexible plastic is not a scientific description. The authors should be more specific in which polymer is comparable!

Response: thank you for your suggestion. We cited this sentence from the literature and didn't realize the non-scientific description. We have added the specific polymers that are comparable, including polyethylene, polyvinyl chloride, and polytetrafluoroethylene.

3. "The collagen secreted on 2 D was random, demonstrating no alignments, however, the collagen secreted on 100 G exhibited orthogonal alignment along the grid fibers (the red lines in Figure 3I showed the504 direction of the fibers). These results indicated that the keratocytes can maintain their phenotype in SF media after culturing for 4 weeks, and the grid scaffold can induce the secretion of orthogonally aligned keratocyte specific ECM."

This is a statement is not supported by a quantitative data set. Yet it is a very important aspect in the regeneration of corneal tissues. The authors have to measure this alignment e.g. via picrosirius red staining of the collagen VI or second harmonic study protocols.

Response: thank you for your suggestion. We tried to stain the collagen VI using the dye of picosirius red, but it would be difficult to perform the quantitative determination using the staining result since the thickness of our construct (over 100 μm) is much larger than the optimal staining thickness (about 6 μm) for this staining method. We need to section the construct to about 6 μm thickness, but the cell distribution is not even within the construct and the collagen secreted by the cells within the 6 μm thickness is not enough for the quantitative determination. Thus, we put forward to characterizing the alignment of collagen VI by measuring the orientation degree of the nuclei of the keratocyte on the grid fibers as the collagen VI is secreted on the cell membrane, the cell growth orientation can also reflect the distribution of the secreted ECM. We measured the deviation degree of the nuclei with respect to the fibers, using the software of Image Pro Plus, from the Collagen VI staining images shown below after cells cultured on the fiber hydrogel for 4 weeks under the serum free media. We measured the degrees of 100 nuclei in each image and calculated the percentage of the cells within a range of degree, including 0-10°, 10-20°, 20-30° and 30-45°, and we assumed that the cells within 0-10° were aligned along the fibers. From the result shown in Figure 3J, we can find out that about 60% cells had aligned growth orientation, which can indicate that most secreted collagen were aligned along the fibers to prove the conclusion we made.

Figure. Immunofluorescent staining images of the collagen VI secreted by LSSCs inoculated on three 100 G scaffolds in SF media after culturing for 4 weeks. Nuclei were stained with DAPI (blue). The scale bar is 100 μm .

Fig. 3 A) Vimentin expression and cytoskeleton staining of LSSCs in SC media on 2 D TCPs, 3 D GelMA, and the 100 G construct after culturing for 7 days. Images show the fluorescent staining of Vimentin (green), phalloidin (red), and nuclei (blue). B) ALDH3A1 expression and cytoskeleton staining of LSSCs in SF media on 2 D TCPs, 3 D GelMA, and the 100 G construct after culturing for 7 days. Images show fluorescent staining of ALDH3A1 (green), phalloidin (red), and nuclei (red). C)-H) The expression of KERATOCAN, ALDH3A1, AQP1, and THY1 in LSSCs cultured on 2 D TCPs, 3 D GelMA, and the 100 G construct were quantified by qPCR after cultured for 2 weeks in SC media or SF media; quantification was normalized by the β -actin signal. Data are expressed as the average \pm S.D. of three independent experiments ($n = 3$). I) Immunofluorescent staining of the collagen VI secreted by LSSCs inoculated on 2 D TCPs or in 100 G in SF media after culturing for 4 weeks. Nuclei were stained with DAPI (blue). The scale bar is 100 μ m. J) The cell alignment degree along the orthogonally aligned fibers on the 100 G construct after 4 weeks culture under the SF media.

4. “The optimal fiber spacing of 100 μ m was verified to create a fiber hydrogel construct with the mechanical properties, light transmittance, and swelling ratio most similar to the native corneal tissue.” It should be clarified that this results are for rat cornea. Human cornea might need other parameters to be optimal” E.g. sentence

could be “The optimal fiber spacing of 100 um was verified to create a fiber hydrogel construct with the mechanical properties, light transmittance, and swelling ratio most similar to the native corneal tissue of rats. Future studies need to verify of this parameters would be the same for human applications.”

Response: We have revised this sentence as you suggested, and it will be more rigorous. Thank you very much for your advice.

Reviewers' Comments:

Reviewer #1:

Remarks to the Author:

the authors did revise the manuscript baed on all the comments of the reviewer.

Comments:

The authors did revise the manuscript based on all the comments of the reviewer.

Dear Reviewer:

Thank you very much for all your useful suggestions and comments to our manuscript.